# Domain Generalization via Nuclear Norm Regularization

Zhenmei Shi*,  Yifei Ming*,  Ying Fan*,  Frederic Sala,  Yingyu Liang
University of Wisconsin-Madison
{zhmeishi,alvinming,yingfan,fredsala,yliang}@cs.wisc.edu

The ability to generalize to unseen domains is crucial for machine learning systems deployed in the real world, especially when we only have data from limited training domains. In this paper, we propose a simple and effective regularization method based on the nuclear norm of the learned features for domain generalization. Intuitively, the proposed regularizer mitigates the impacts of environmental features and encourages learning domain-invariant features. Theoretically, we provide insights into why nuclear norm regularization is more effective compared to ERM and alternative regularization methods. Empirically, we conduct extensive experiments on both synthetic and real datasets. We show nuclear norm regularization achieves strong performance compared to baselines in a wide range of domain generalization tasks. Moreover, our regularizer is broadly applicable with various methods such as ERM and SWAD with consistently improved performance, e.g., 1.7% and 0.9% test accuracy improvements respectively on the DomainBed benchmark.

## 1. Introduction

Making machine learning models reliable under distributional shifts is crucial for real-world applications such as autonomous driving, health risk prediction, and medical imaging. This motivates the area of domain generalization, which aims to obtain models that generalize to unseen domains, e.g., different image backgrounds or different image styles, by learning from a limited set of training domains. To improve model robustness under domain shifts, a plethora of algorithms have been recently proposed [1–5]. In particular, methods that learn invariant feature representations (class-relevant patterns) or invariant predictors [6] across domains demonstrate promising performance both empirically and theoretically [7–10]. Despite this, it remains challenging to improve on empirical risk minimization (ERM) when evaluating a broad range of real-world datasets [11, 12]. Notice that ERM is a reasonable baseline method since it must use invariant features to achieve optimal in-distribution performance. It has been empirically shown [13] that ERM already learns "invariant" features sufficient for domain generalization, which means these features are only correlated with the class label, not domains or environments.

Although competitive in domain generalization tasks, the main issue ERM faces is that the invariant features it learns can be arbitrarily mixed: environmental features are hard to disentangle from invariant features. Various regularization techniques that control empirical risks across domains have been proposed [6, 14, 15], but few directly regularize ERM, motivating this work. One desired property to improve ERM is disentangling the invariant features from the mixtures. As low-dimensional structures prevail in deep learning, a natural way to achieve this is to identify the subset of solutions from ERM with minimal information retrieved from training domains by controlling the rank. This parsimonious method may avoid domain overfitting. We are interested in the following question:

*Can ERM benefit from rank regularization of the extracted feature for better domain generalization?*

To answer this question, we propose a simple yet effective algorithm, ERM-NU (Empirical Risk Minimization with Nuclear Norm Regularization), for improving domain generalization without

---

*Equal contribution.

First Conference on Parsimony and Learning (CPAL 2024).

acquiring domain annotations. Our method is inspired by works in low-rank matrix completion and recovery with nuclear norm minimization [16–21]. Given feature representations from pre-trained models via ERM, ERM-NU aims to extract class-relevant (domain-invariant) features and to rule out spurious (environmental) features by fine-tuning the network with nuclear norm regularization. Specifically, we propose to minimize the nuclear norm of the backbone features, which is a convex envelope to the rank of the feature matrix [22].

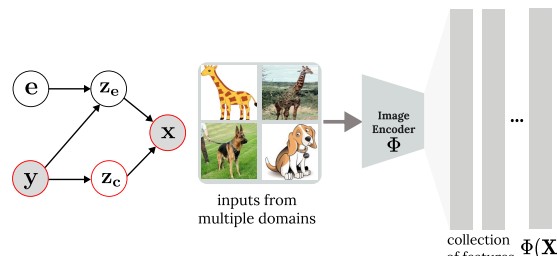 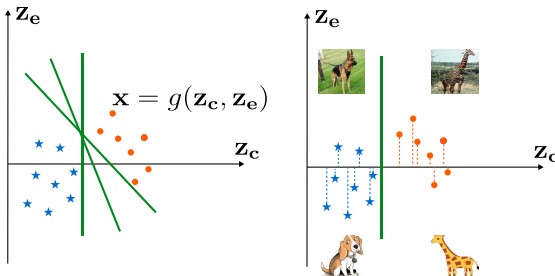

(a) Left: Causal graph inspired by [15]. Shaded variables are observed. Right: Training pipeline. We collect input features from multiple domains.

(b) Left: ERM Solutions (green lines). Right: ERM solution with the smallest nuclear norm of extracted feature ($\mathbf{z_c}$ only).

Figure 1: Causal graph of our data assumption (1a), and the effect of nuclear norm regularization in ERM (1b) where we use a linear $g$ for a simple illustration. In Figure 1a, $\mathbf{z}_e$ has a spurious correlation to $y$, while $\mathbf{z}_c$ only depends on $y$. From Figure 1b, nuclear norm regularization can select a subset of ERM solutions that extract the smallest possible information (in the sense of rank) from $\mathbf{x}$ for classification, which can reduce the effect of environmental features for better generalization performance while still preserving high classification accuracy.

Our main contributions and findings are as follows:

- **ERM-NU offers competitive empirical performance**: We evaluate the performance of ERM-NU on synthetic datasets and five benchmark real-world datasets. Despite its simplicity, NU demonstrates strong performance and improves on existing methods on some large-scale datasets such as TerraInc and DomainNet.
- **We provide theoretical insights** when applying ERM-NU to domain generalization tasks: We show that even training with infinite data from in-domain (ID) tasks on a specific data distribution, ERM with weight decay may perform worse than random guessing on out-of-domain (OOD) tasks, while ERM with bounded rank (corresponding to ERM-NU) can guarantee 100% test accuracy on the out-of-domain task.
- **Nuclear norm regularization (NU) is simple, efficient, and broadly applicable**: NU is computationally efficient as it does not require annotations from training domains. As a regularization, NU is also potentially orthogonal to other methods that are based on ERM: we get a consistent improvement of NU on ERM, Mixup [23] and SWAD [24] as baselines.

## 2. Method

### 2.1. Preliminaries

We use $\mathcal{X}$ and $\mathcal{Y}$ to denote the input and label space, respectively. Following [5, 12, 15], we consider data distributions consisting of environments (domains) $\mathcal{E} = \{1, \dots, E\}$. For a given environment $e \in \mathcal{E}$ and label $y \in \mathcal{Y}$, the data generation process is the following: latent *environmental/spurious* features (e.g., image style or background information) $\mathbf{z}_e$ and *invariant/class-relevant* features (e.g., windows pattern for house images) $\mathbf{z}_c$ are sampled where invariant features only depend on $y$, while environmental features depend on $e$ and $y$ (i.e., environmental features and the label may have spurious correlations), $\mathbf{z}_c \perp \mathbf{z}_e$. The input data is generated from the latent features $\mathbf{x} = g(\mathbf{z}_c, \mathbf{z}_e)$ by some injective function $g$. See illustration in Figure 1a. We assume that the training data is drawn from a mixture of $E^{tr} \subset \mathcal{E}$ domains and test data is drawn from some unseen domain in $E^{ts} \subset \mathcal{E}$. In the domain shift setup, training domains are disjoint from test domains: $\mathcal{E}^{tr} \cap \mathcal{E}^{ts} = \emptyset$. In this

work, as we do not require domain annotations for training data, we remove notation involving $\mathcal{E}$ for simplicity and denote the training data distribution as $\mathcal{D}_{\text{id}}$ and the unseen domain test data distribution as $\mathcal{D}_{\text{ood}}$. We consider population risk. Our objective is to learn a feature extractor $\Phi :$ $\mathcal{X} \to \mathbb{R}^d$ that maps input data to a $d$-dimensional feature embedding (usually fine-tuned from a pre-trained backbone, e.g. ResNet [25] pre-trained on ImageNet) and a classifier $\hat{f}$ to minimize the risk on *unseen* environments, $\mathcal{L}(\hat{f}, \Phi) := \mathbb{E}_{(\mathbf{x},y)\sim\mathcal{D}_{\text{ood}}} \left[ \ell(\hat{f}(\Phi(\mathbf{x})), y) \right]$, where the function $\ell$ can be any loss appropriate to classification, e.g., cross-entropy. The *nuclear/trace norm* [26] of a matrix is the sum of the singular values of the matrix. Suppose a matrix $\mathbf{M} \in \mathbb{R}^{m \times n}$, we have the nuclear norm

$$\|\mathbf{M}\|_* := \sum_i^{\min\{m,n\}} \sigma_i(\mathbf{M}),$$

where $\sigma_i(\mathbf{M})$ is the $i$-th largest singular value. From [22], we know the nuclear norm is the tightest convex envelope of the rank function of a matrix within the unit ball, i.e., the nuclear norm is smaller than the rank when the operator norm (spectral norm) $\|\mathbf{M}\|_2 = \sigma_1(\mathbf{M}) \le 1$. As the matrix rank function is highly non-convex, nuclear norm regularization is often used in optimization to achieve a low-rank solution, as it has good convergence guarantees, while the rank function does not.

## 2.2. Method description

**Intuition.** Intuitively, to guarantee low risk on $\mathcal{D}_{\text{ood}}$, $\Phi$ needs to rely only on invariant features for prediction. It must not use environmental features in order to avoid spurious correlations to ensure domain generalization. As environmental features depend on the label $y$ and the environment $e$ in Figure 1a, our main hypothesis is that *environmental features have a lower correlation with the label than the invariant features*. If our hypothesis is true, we can eliminate environmental features by constraining the rank of the learned representations from the training data while minimizing the empirical risk, i.e., the invariant features will be preserved (due to empirical risk minimization) and the environmental features will be removed (due to rank minimization).

**Objectives.** We consider fine-tuning the backbone (feature extractor) $\Phi$ with a linear prediction head. Denote the linear head as $\mathbf{a} \in \mathbb{R}^{d \times m}$, where $m$ is the class number. The goal of ERM is to minimize the expected risk $\mathcal{L}(\mathbf{a}, \Phi) := \mathbb{E}_{(\mathbf{x},y)\sim\mathcal{D}_{\text{in}}} \left[ \ell(\mathbf{a}^\top \Phi(\mathbf{x}), y) \right]$.

Consider the latent vector $\Phi(\mathbf{x}) \in \mathbb{R}^d$. This vector may contain both environment-related and class-relevant features. In order to obtain just the class-relevant features, we would like for $\Phi$ to extract as little information as possible while simultaneously optimizing the ERM loss. See illustration in Figure 1b. Note that, we assume that the correlation between environmental features and labels is lower than the correlation between invariant features and labels. Let $\mathbf{X}$ be a batch of training data points (batch size $> d$). To minimize information and so rule out environmental features, we minimize the rank of $\Phi(\mathbf{X})$. Our objective is

$$\min_{\mathbf{a}, \Phi} \mathcal{L}(\mathbf{a}, \Phi) + \lambda \text{rank}(\Phi(\mathbf{X})). \tag{1}$$

As the nuclear norm is a convex envelope to the rank of a matrix, our convex relaxation objective is

$$\min_{\mathbf{a}, \Phi} \mathcal{L}(\mathbf{a}, \Phi) + \lambda \|\Phi(\mathbf{X})\|_*, \tag{2}$$

where $\lambda$ is the regularization weight. Finally, we use Equation (2) as our main loss function.

**Takeaways.** We summarize the advantages of nuclear norm minimization as follows:

- Simple and efficient: Our method can be easily implemented, e.g., NU only needs two more lines of code as shown below. Also, the model inference speed doesn't change after training.
- Broadly applicable: Without requiring domain labels, our method can be used in conjunction with a broad range of existing domain generalization algorithms.
- Empirically effective and theoretically sound: Our method demonstrates promising performance on synthetic and real-world tasks (Section 3) with theoretical insights (Section 4).

```
1  def forward(self, x, y):
2      f = self.featurizer(x) # get feature embedding
3      loss = F.cross_entropy(self.classifier(f), y) # get classification loss
4      _,s,_ = torch.svd(f) # singular value decomposition
5      loss += self.lambda * torch.sum(s) # add nuclear norm regularization
6      return loss
```

## 3. Experiments

In this section, we start by presenting a synthetic task in Section 3.1 to help visualize the effects of nuclear norm regularization. Next, in Section 3.2, we demonstrate the effectiveness of our approach with real-world datasets. We provide further discussions and ablation studies in Section 3.3.

### 3.1. Synthetic tasks

To visualize the effects of nuclear norm regularization, we start with a synthetic dataset with binary labels and two-dimensional inputs. Our expectation is that un-regularized ERM will perform well for in-domain (ID) data but struggle for out-of-domain (OOD) data, whereas nuclear norm-regularized ERM will excel in both settings. Assume inputs $\mathbf{x} = [\mathbf{x}_1, \mathbf{x}_2]$, where $\mathbf{x}_1$ is the invariant feature and $\mathbf{x}_2$ is the environmental feature. Specifically, $\mathbf{x}_1$ is drawn from a uniform distribution conditioned on $y \in \{-1, 1\}$ for both ID (training) and OOD (test) datasets:

$$\mathbf{x}_1 \mid y = 1 \sim \mathcal{U}[0, 1], \quad \mathbf{x}_1 \mid y = -1 \sim \mathcal{U}[-1, 0]$$

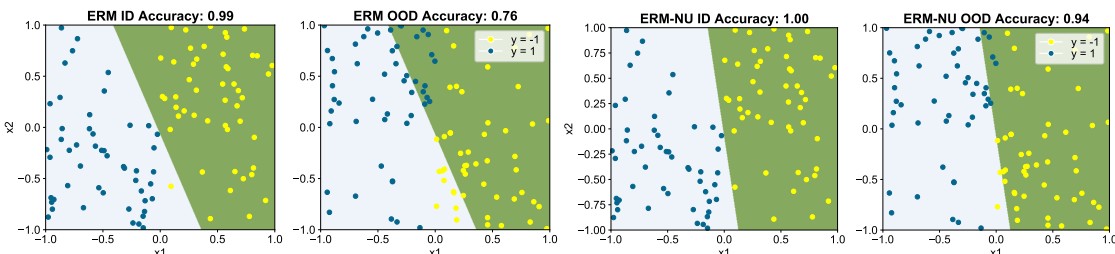

Figure 2: ID and OOD classification results with ERM and ERM-NU on the synthetic dataset with two classes (shown in yellow and navy blue). We visualize the decision boundary. While the model achieves nearly perfect accuracy on ID training set, the performance drastically degrades on the OOD test set. Nuclear norm regularization significantly reduces the OOD error rate.

The environmental feature $\mathbf{x}_2$ also follows a uniform distribution but is conditioned on both $y$ and a Bernoulli random variable $b \sim \text{Ber}(0.7)$. For ID data, $\mathbf{x}_2 \mid y = 1 \sim \mathcal{U}[0, 1]$ with probability (w.p.) 0.7, while $\mathbf{x}_2 \mid y = 1 \sim \mathcal{U}[-1, 0]$ w.p. 0.3. In contrast, for OOD data, $\mathbf{x}_2 \mid y = 1 \sim \mathcal{U}[-1, 0]$ w.p. 0.7, and $\mathbf{x}_2 \mid y = 1 \sim \mathcal{U}[0, 1]$ w.p. 0.3. We provide a more general setting in Section 4.

We visualize the ID and OOD datasets in Figure 2, where samples from $y = -1$ and $y = 1$ are shown in yellow and navy blue dots, respectively. We consider a simple linear feature extractor $\Phi(\mathbf{x}) = A\mathbf{x}$ with $A \in \mathbb{R}^{2 \times 2}$. The models are trained with ERM and ERM-NU objectives using gradient descent until convergence. To better illustrate the effects of nuclear norm minimization, we show the decision boundary along with the accuracy on ID and OOD datasets. For ID dataset, training with both objective yield nearly perfect accuracy. For OOD dataset, the model trained with ERM only achieves an accuracy of 0.76, as a result of utilizing the environmental feature. In contrast, training with ERM-NU successfully mitigates the reliance on environmental features and significantly improves the OOD accuracy to 0.94. In Section 4, we further provide theoretical analysis to better understand the effects of nuclear norm regularization.

### 3.2. Real-world tasks

We demonstrate the effects of nuclear norm regularization across real-world datasets and compare them with a broad range of algorithms.

| Algorithm | VLCS | PACS | OfficeHome | TerraInc | DomainNet | Average |
|---|---|---|---|---|---|---|
| MMD[†] (CVPR 18) [10] | 77.5 ± 0.9 | 84.6 ± 0.5 | 66.3 ± 0.1 | 42.2 ± 1.6 | 23.4 ± 9.5 | 58.8 |
| Mixstyle[‡] (ICLR 21) [27] | 77.9 ± 0.5 | 85.2 ± 0.3 | 60.4 ± 0.3 | 44.0 ± 0.7 | 34.0 ± 0.1 | 60.3 |
| GroupDRO[†] (ICLR 19) [28] | 76.7 ± 0.6 | 84.4 ± 0.8 | 66.0 ± 0.7 | 43.2 ± 1.1 | 33.3 ± 0.2 | 60.7 |
| IRM[†] (ArXiv 20) [6] | 78.5 ± 0.5 | 83.5 ± 0.8 | 64.3 ± 2.2 | 47.6 ± 0.8 | 33.9 ± 2.8 | 61.6 |
| ARM[†] (ArXiv 20) [29] | 77.6 ± 0.3 | 85.1 ± 0.4 | 64.8 ± 0.3 | 45.5 ± 0.3 | 35.5 ± 0.2 | 61.7 |
| VREx[†] (ICML 21) [14] | 78.3 ± 0.2 | 84.9 ± 0.6 | 66.4 ± 0.6 | 46.4 ± 0.6 | 33.6 ± 2.9 | 61.9 |
| CDANN[†] (ECCV 18) [8] | 77.5 ± 0.1 | 82.6 ± 0.9 | 65.8 ± 1.3 | 45.8 ± 1.6 | 38.3 ± 0.3 | 62.0 |
| AND-mask[*] (ICLR 20)[30] | 78.1 ± 0.9 | 84.4 ± 0.9 | 65.6 ± 0.4 | 44.6 ± 0.3 | 37.2 ± 0.6 | 62.0 |
| DANN[†] (JMLR 16) [7] | 78.6 ± 0.4 | 83.6 ± 0.4 | 65.9 ± 0.6 | 46.7 ± 0.5 | 38.3 ± 0.1 | 62.6 |
| RSC[†] (ECCV 20) [31] | 77.1 ± 0.5 | 85.2 ± 0.9 | 65.5 ± 0.9 | 46.6 ± 1.0 | 38.9 ± 0.5 | 62.7 |
| MTL[†] (JMLR 21) [32] | 77.2 ± 0.4 | 84.6 ± 0.5 | 66.4 ± 0.5 | 45.6 ± 1.2 | 40.6 ± 0.1 | 62.9 |
| Mixup[†] (ICLR 18) [1] | 77.4 ± 0.6 | 84.6 ± 0.6 | 68.1 ± 0.3 | 47.9 ± 0.8 | 39.2 ± 0.1 | 63.4 |
| MLDG[†] (AAAI 18) [33] | 77.2 ± 0.4 | 84.9 ± 1.0 | 66.8 ± 0.6 | 47.7 ± 0.9 | 41.2 ± 0.1 | 63.6 |
| Fish (ICLR 22) [34] | 77.8 ± 0.3 | 85.5 ± 0.3 | 68.6 ± 0.4 | 45.1 ± 1.3 | 42.7 ± 0.2 | 63.9 |
| Fishr[*] (ICML 22) [35] | 77.8 ± 0.1 | 85.5 ± 0.4 | 67.8 ± 0.1 | 47.4 ± 1.6 | 41.7 ± 0.0 | 64.0 |
| SagNet[†] (CVPR 21) [36] | 77.8 ± 0.5 | 86.3 ± 0.2 | 68.1 ± 0.1 | 48.6 ± 1.0 | 40.3 ± 0.1 | 64.2 |
| SelfReg (ICCV 21) [37] | 77.8 ± 0.9 | 85.6 ± 0.4 | 67.9 ± 0.7 | 47.0 ± 0.3 | 41.5 ± 0.2 | 64.2 |
| CORAL[†] (ECCV 16) [9] | 78.8 ± 0.6 | 86.2 ± 0.3 | 68.7 ± 0.3 | 47.6 ± 1.0 | 41.5 ± 0.1 | 64.5 |
| SAM[‡] (ICLR 21) [38] | 79.4 ± 0.1 | 85.8 ± 0.2 | 69.6 ± 0.1 | 43.3 ± 0.7 | 44.3 ± 0.0 | 64.5 |
| mDSDI (NeurIPS 21) [39] | 79.0 ± 0.3 | 86.2 ± 0.2 | 69.2 ± 0.4 | 48.1 ± 1.4 | 42.8 ± 0.1 | 65.1 |
| MIRO (ECCV 22) [40] | 79.0 ± 0.0 | 85.4 ± 0.4 | 70.5 ± 0.4 | 50.4 ± 1.1 | 44.3 ± 0.2 | 65.9 |
| ERM[†] [41] | 77.5 ± 0.4 | 85.5 ± 0.2 | 66.5 ± 0.3 | 46.1 ± 1.8 | 40.9 ± 0.1 | 63.3 |
| **ERM-NU (ours)** | **78.3 ± 0.3** | **85.6 ± 0.1** | **68.1 ± 0.1** | **49.6 ± 0.6** | **43.4 ± 0.1** | **65.0** |
| SWAD[‡] (NeurIPS 21) [24] | 79.1 ± 0.1 | 88.1 ± 0.1 | 70.6 ± 0.2 | 50.0 ± 0.3 | 46.5 ± 0.1 | 66.9 |
| **SWAD-NU (ours)** | **79.8 ± 0.2** | **88.5 ± 0.2** | **71.3 ± 0.3** | **52.2 ± 0.3** | **47.1 ± 0.1** | **67.8** |

Table 1: OOD accuracy for five realistic domain generalization datasets. The results marked by [†], [‡], [*] are the reported numbers from [11], [24], [35] respectively. We highlight **our methods** in bold. The results of Fish, SelfReg, mDSDI and MIRO are the reported ones from each paper. Average accuracy and standard errors are reported from three trials. Nuclear norm regularization is simple, effective, and broadly applicable. It significantly improves the performance over ERM and a competitive baseline SWAD across all datasets considered.

**Experimental setup.** Nuclear norm regularization is simple, flexible, and can be plugged into ERM-like algorithms. To verify its effectiveness, we consider adding the regularizer over ERM and SWAD (dubbed as ERM-NU and SWAD-NU, respectively). For a fair comparison with baseline methods, we evaluate our algorithm on the DomainBed testbed [11], an open-source benchmark that aims to rigorously compare different algorithms for domain generalization. The testbed consists of a wide range of datasets for multi-domain image classification tasks, including PACS [42] (4 domains, 7 classes, 9,991 images), VLCS [43] (4 domains, 5 classes, 10,729 images), Office-Home [44] (4 domains, 65 classes, 15,500 images), Terra Incognita [45] (4 domains, 10 classes, 24,788 images), and DomainNet [46] (6 domains, 345 classes, 586,575 images).

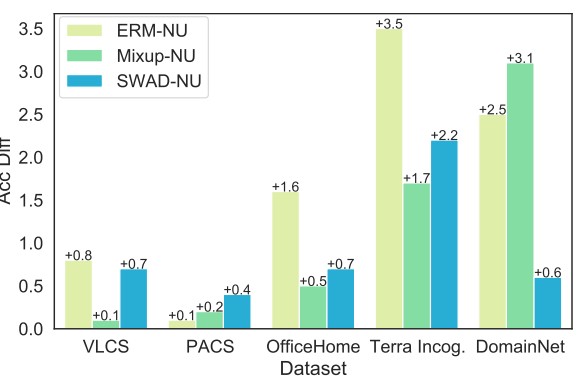

Figure 3: Nuclear norm regularization enhances competitive baselines across a range of realistic datasets, as demonstrated by the average difference in accuracy for ERM, Mixup, and SWAD. Detailed results for individual datasets can be seen in Table 2.

Following the evaluation protocol in DomainBed, we report all performance scores by "leave-one-out cross-validation", where averaging over cases that use one domain as the test (OOD) domain and all others as the training (ID) domains. For the model selection criterion, we use the "training-domain validation set" strategy, which refers to choosing the model maximizing the accuracy on the overall validation set, 20% of training domain data. For each dataset and model, we report the test domain accuracy of the best-selected model (average over three independent runs with different random seeds). Following common practice, we use ResNet-50 [25] as the feature backbone. We use the output features of the penultimate layer of ResNet-50 for nu-

clear norm regularization and fine-tune the whole model. The default value of weight scale $\lambda$ is set as $0.01$ and distributions for random search as $10^{\text{Uniform}\,(-2.5,-1.5)}$. The default batch size is 32 and the distribution for random search is $2^{\text{Uniform}\,(5,6)}$. During training, we perform batch-wise nuclear norm regularization, similar to [6] which uses batch-wise statistics for invariant risk minimization.

**Nuclear norm regularization achieves strong performance across a wide range of datasets.** We present an overview of the OOD accuracy for DomainBed datasets across various algorithms in Table 1. We observe that: (1) incorporating nuclear norm regularization consistently improves the performance of ERM and SWAD across all datasets considered. In particular, compared to ERM, ERM-NU yields an average accuracy improvement of 1.7%. (2) SWAD-NU demonstrates highly competitive performance relative to other baselines, including prior invariance-learning approaches such as IRM, VREx, and DANN. Notably, the approach does not require domain labels, which further underscores the versatility of nuclear norm regularization for real-world datasets.

**Nuclear norm regularization significantly improves baselines.** Across a range of realistic datasets, nuclear norm regularization enhances competitive baselines. To examine whether NU is effective with baselines other than SWAD, in Figure 3, we plot the average difference in accuracy with and without nuclear norm regularization for ERM, Mixup, and SWAD. Detailed results for individual datasets are in Table 2. Encouragingly, adding nuclear norm regularization improves the performance over all three baselines across the five datasets. In particular, the average accuracy is improved by 3.5 with ERM-NU over ERM on Terra Incognita, and 3.1 with Mixup-NU over Mixup on DomainNet. This further suggests the effectiveness of nuclear norm regularization in learning invariant features. See full results in Appendix E.

| Algorithm | C | L | S | V | Average |
|---|---|---|---|---|---|
| ERM | 97.7 ± 0.4 | 64.3 ± 0.9 | 73.4 ± 0.5 | 74.6 ± 1.3 | 77.5 |
| ERM-NU | 97.9 ± 0.4 | 65.1 ± 0.3 | 73.2 ± 0.9 | 76.9 ± 0.5 | **78.3** |
| Mixup | 98.3 ± 0.6 | 64.8 ± 1.0 | 72.1 ± 0.5 | 74.3 ± 0.8 | 77.4 |
| Mixup-NU | 97.9 ± 0.2 | 64.1 ± 1.4 | 73.1 ± 0.9 | 74.8 ± 0.5 | **77.5** |
| SWAD | 98.8 ± 0.1 | 63.3 ± 0.3 | 75.3 ± 0.5 | 79.2 ± 0.6 | 79.1 |
| SWAD-NU | 99.1 ± 0.4 | 63.6 ± 0.4 | 75.9 ± 0.4 | 80.5 ± 1.0 | **79.8** |

(a) VLCS

| Algorithm | A | C | P | S | Average |
|---|---|---|---|---|---|
| ERM | 84.7 ± 0.4 | 80.8 ± 0.6 | 97.2 ± 0.3 | 79.3 ± 1.0 | 85.5 |
| ERM-NU | 87.4 ± 0.5 | 79.6 ± 0.9 | 96.3 ± 0.7 | 79.0 ± 0.5 | **85.6** |
| Mixup | 86.1 ± 0.5 | 78.9 ± 0.8 | 97.6 ± 0.1 | 75.8 ± 1.8 | 84.6 |
| Mixup-NU | 86.7 ± 0.3 | 78.0 ± 1.3 | 97.3 ± 0.3 | 77.3 ± 2.0 | **84.8** |
| SWAD | 89.3 ± 0.2 | 83.4 ± 0.6 | 97.3 ± 0.3 | 82.5 ± 0.5 | 88.1 |
| SWAD-NU | 89.8 ± 1.1 | 82.8 ± 1.0 | 97.7 ± 0.3 | 83.7 ± 1.1 | **88.5** |

(b) PACS

| Algorithm | A | C | P | R | Average |
|---|---|---|---|---|---|
| ERM | 61.3 ± 0.7 | 52.4 ± 0.3 | 75.8 ± 0.1 | 76.6 ± 0.3 | 66.5 |
| ERM-NU | 63.3 ± 0.2 | 54.2 ± 0.3 | 76.7 ± 0.2 | 78.2 ± 0.3 | **68.1** |
| Mixup | 62.4 ± 0.8 | 54.8 ± 0.6 | 76.9 ± 0.3 | 78.3 ± 0.2 | 68.1 |
| Mixup-NU | 64.3 ± 0.5 | 55.9 ± 0.6 | 76.9 ± 0.4 | 78.0 ± 0.6 | **68.8** |
| SWAD | 66.1 ± 0.4 | 57.7 ± 0.4 | 78.4 ± 0.1 | 80.2 ± 0.2 | 70.6 |
| SWAD-NU | 67.5 ± 0.3 | 58.4 ± 0.6 | 78.6 ± 0.9 | 80.7 ± 0.1 | **71.3** |

(c) OfficeHome

| Algorithm | L100 | L38 | L43 | L46 | Average |
|---|---|---|---|---|---|
| ERM | 49.8 ± 4.4 | 42.1 ± 1.4 | 56.9 ± 1.8 | 35.7 ± 3.9 | 46.1 |
| ERM-NU | 52.5 ± 1.2 | 45.0 ± 0.5 | 60.2 ± 0.2 | 40.7 ± 1.0 | **49.6** |
| Mixup | 59.6 ± 2.0 | 42.2 ± 1.4 | 55.9 ± 0.8 | 33.9 ± 1.4 | 47.9 |
| Mixup-NU | 55.1 ± 3.1 | 45.8 ± 0.7 | 56.4 ± 1.2 | 41.1 ± 0.6 | **49.6** |
| SWAD | 55.4 ± 0.0 | 44.9 ± 1.1 | 59.7 ± 0.4 | 39.9 ± 0.2 | 50.0 |
| SWAD-NU | 58.1 ± 3.3 | 47.7 ± 1.6 | 60.5 ± 0.8 | 42.3 ± 0.9 | **52.2** |

(d) Terra Incognita

| Algorithm | clip | info | paint | quick | real | sketch | Average |
|---|---|---|---|---|---|---|---|
| ERM | 58.1 ± 0.3 | 18.8 ± 0.3 | 46.7 ± 0.3 | 12.2 ± 0.4 | 59.6 ± 0.1 | 49.8 ± 0.4 | 40.9 |
| ERM-NU | 60.9 ± 0.0 | 21.1 ± 0.2 | 49.9 ± 0.3 | 13.7 ± 0.2 | 62.5 ± 0.2 | 52.5 ± 0.4 | **43.4** |
| Mixup | 55.7 ± 0.3 | 18.5 ± 0.5 | 44.3 ± 0.5 | 12.5 ± 0.4 | 55.8 ± 0.3 | 48.2 ± 0.5 | 39.2 |
| Mixup-NU | 59.5 ± 0.3 | 20.5 ± 0.1 | 49.3 ± 0.4 | 13.3 ± 0.5 | 59.6 ± 0.3 | 51.5 ± 0.2 | **42.3** |
| SWAD | 66.0 ± 0.1 | 22.4 ± 0.3 | 53.5 ± 0.1 | 16.1 ± 0.2 | 65.8 ± 0.4 | 55.5 ± 0.3 | 46.5 |
| SWAD-NU | 66.6 ± 0.2 | 23.2 ± 0.2 | 54.3 ± 0.2 | 16.2 ± 0.2 | 66.1 ± 0.6 | 56.2 ± 0.2 | **47.1** |

(e) DomainNet

Table 2: Nuclear norm regularization improves the domain generalization performance over various baselines such as ERM, Mixup, and SWAD.

### 3.3. Ablations and discussions

**Analyzing the regularization strength with stable rank.** We aim to better understand the strength of nuclear norm regularization (used in fine-tuning only) on OOD accuracy. Due to the precision of floating point numbers and numerical perturbation, it is common to use stable rank (numerical rank) to approximate the matrix rank in numerical analysis. Suppose a matrix $\mathbf{M} \in \mathbb{R}^{m \times n}$, the stable rank is defined as: $\text{StableRank}(\mathbf{M}) := \frac{\|\mathbf{M}\|_F^2}{\|\mathbf{M}\|_2^2}$, where $\|\|_2$ is the operator norm (spectral norm) and $\|\|_F$ is the Frobenius norm. The stable rank is analogous to the classical rank of a matrix but considerably more well-behaved. For example, the stable rank is a continuous and Lipschitz function while the rank function is discrete. In Figure 4, we calculate the stable rank of the OOD data feature representation of the ERM-NU model trained with different nuclear norm regularization weight $\lambda$ and we plot the OOD accuracy simultaneously. We have three observations. (1) The stable rank is

decreasing when we have a stronger nuclear norm regularizer, which is consistent with our method intuition. (2) As the nuclear norm regularization weight increases, the OOD accuracy will increase first and then decrease. In the first stage, as we increase nuclear norm regularization weight, the environmental features start to be ruled out and the OOD accuracy improves. In the second stage, when nuclear norm regularization strength is large enough, some invariant features will be ruled out, which will hurt the generalization. (3) Although the ResNet-50 is a 2048-dim feature extractor, the stable rank of the OOD data feature representation is pretty low, e.g, on average the stable rank is smaller than 100. On the other hand, when the dataset becomes more "complicated", the stable rank will increase, e.g., when $\lambda = 0.0$, the stable rank of DomainNet features (6 domains, 345 classes) is over 100, while the stable rank of VLCS (4 domains, 5 classes) features is only around 50.

**Exploring alternative regularizers.** We use SWAD [24] as a baseline, which aims to find flat minima that suffers less from overfitting by a dense and overfit-aware stochastic weight sampling strategy. We consider different regularizers with SWAD: CORAL [9] tries to minimize

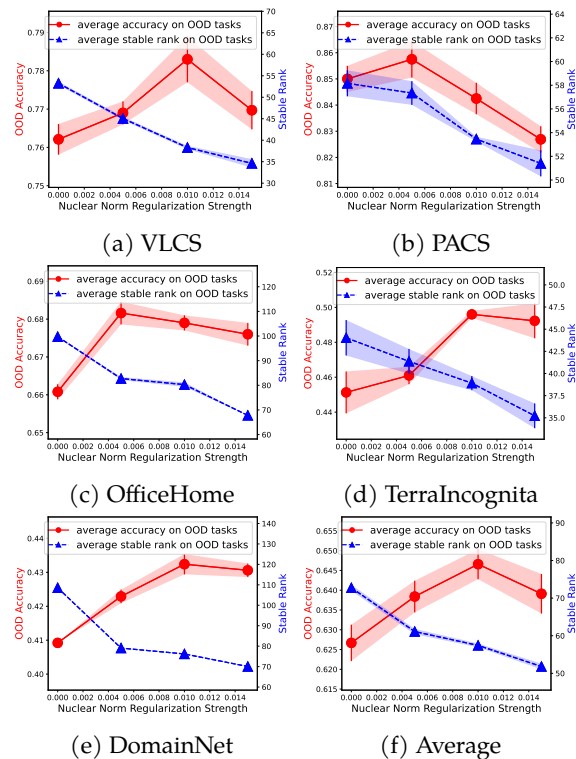

Figure 4: Stable rank and OOD accuracy of ERM-NU with varying nuclear norm regularization weight $\lambda$ ($x$-axis) on different datasets.

domain shift by aligning the second-order statistics of input data from training and test domains. MIRO [40], one of the SOTA regularization methods in domain generalization, uses mutual information to reduce the distance between the pre-training model and the fine-tuned model. The performance comparison is shown in Table 3, where we observe that nuclear norm regularization consistently achieves competitive performance compared to alternative regularizers.

## 4. Theoretical Analysis

Next, we present a simple but insightful theoretical result showing that, for a more general setting defined in Section 3.1, the ERM-rank solution to the Equation (1) is much more robust than the ERM solution on OOD tasks.

**Data distributions.** We consider the binary classification setting for simplification [6, 15].

| Algorithm | VLCS | PACS | DomainNet | Average |
|---|---|---|---|---|
| SWAD [24] | $79.1 \pm 0.1$ | $88.1 \pm 0.1$ | $46.5 \pm 0.1$ | 71.2 |
| SWAD-CORAL [9] | $78.9 \pm 0.1$ | $88.3 \pm 0.1$ | $46.8 \pm 0.0$ | 71.3 |
| SWAD-MIRO [40] | $79.6 \pm 0.2$ | $88.4 \pm 0.1$ | $47.0 \pm 0.0$ | 71.7 |
| SWAD-NU (ours) | $\textbf{79.8} \pm 0.2$ | $\textbf{88.5} \pm 0.2$ | $\textbf{47.1} \pm 0.1$ | **71.8** |

Table 3: Alternative regularizers with SWAD on the DomainBed benchmark. Full Table is in Appendix E.

Let $\mathcal{X}$ be the input space, and $\mathcal{Y} = \{\pm 1\}$ be the label space. Let $\tilde{\mathbf{z}} : \mathcal{X} \rightarrow \mathbb{R}^d$ be a feature pattern encoder of the input data $\mathbf{x}$, i.e., $\tilde{\mathbf{z}}(\mathbf{x}) \in \mathbb{R}^d$. For any $j \in [d]$, we see $\tilde{\mathbf{z}}_j$ is a specific feature pattern encoder, i.e $\tilde{\mathbf{z}}_j(\mathbf{x})$ being the $j$-th dimension of $\tilde{\mathbf{z}}(\mathbf{x})$. Suppose $\mathbf{x}$ are drawn from some distribution condition on the label $y$, then we have $\tilde{\mathbf{z}}(\mathbf{x})$ drawn from some distribution condition on the label $y$. For simplicity, we denote $\mathbf{z} = \tilde{\mathbf{z}}(\mathbf{x})y$. We assume, for any $j, j' \in [d]$, $\mathbf{z}_j$ and $\mathbf{z}_{j'}$ are independent condition on $y$ when $j \neq j'$. Let $R \subseteq [d]$ be a subset of size $r$ corresponding to the invariant features, and let $U = [d] \setminus R$ be a subset of size $d - r$ corresponding to the spurious features. With slight abuse of notation, if $\tilde{\mathbf{z}} = g^{-1}$, $y\mathbf{z}_R$ and $y\mathbf{z}_U$ correspond to $\mathbf{z}_c$ and $\mathbf{z}_e$ in Figure 1a respectively.

Next, we define ID tasks and OOD tasks. We inject all the randomness into $\mathbf{z}$. For invariant features in both ID and OOD tasks, we assume that for any $j \in R$, $\mathbf{z}_j \sim [0, 1]$ uniformly, so $\mathbb{E}[\mathbf{z}_j] = \frac{1}{2}$.

Then, we define $\mathcal{D}_\gamma$. A random variable $z \sim \mathcal{D}_\gamma$ means, $z \sim [0,1]$ uniformly with probability $\frac{1}{2} + \gamma$ and $z \sim [-1,0]$ uniformly with probability $\frac{1}{2} - \gamma$, so $\mathbb{E}[z] = \gamma$. In ID tasks, for any environmental features $j \in U$, we assume that $\mathbf{z}_j \sim \mathcal{D}_\gamma$, where $\gamma \in \left(\frac{3}{\sqrt{r}}, \frac{1}{2}\right)$. In OOD tasks, for any $j \in U$, we assume that $\mathbf{z}_j \sim \mathcal{D}_{-\gamma}$. We denote the corresponding distributions as $\mathcal{D}_{\mathrm{id}}$ and $\mathcal{D}_{\mathrm{ood}}$ respectively. We note that the difference between $\mathcal{D}_{\mathrm{id}}$ and $\mathcal{D}_{\mathrm{ood}}$ is that the environmental features have different spurious correlations with label $y$, i.e., different $e$ in Figure 1a.

**Explanation and intuition for our data distributions.** There is an upper bound for $\gamma$ because the environmental features have a smaller correlation with the label than the invariant features, e.g., when $\gamma = \frac{1}{2}$ we cannot distinguish invariant features and environmental features. We also have a lower bound for $\gamma$ to distinguish environmental features and noise. When $\gamma = 0$, the ID task and the OOD task will be identical (no distribution shift). We can somehow use $\gamma$ to measure the "distance" between the ID task and the OOD task. The intuition about the definition of $\mathcal{D}_{\mathrm{id}}$ and $\mathcal{D}_{\mathrm{ood}}$ is that the environmental features may have different spurious correlations with labels in different tasks, while the invariant features keep the same correlations with labels through different tasks.

**Objectives.** For any $j \in [d]$, we assume the feature embedder (defined in Section 2, see Figure 1a) $\Phi(\mathbf{x})_j = \mathbf{w}_j \tilde{\mathbf{z}}_j(\mathbf{x})$ where $\tilde{\mathbf{z}}_j$ is a specific feature pattern encoder and $\mathbf{w}_j$ is a scalar, i.e, the strength of the corresponding feature pattern encoder. We simplify the fine-tuning process, setting $\mathbf{a} = [1, 1, \ldots, 1]^\top$ and the trainable parameter to be $\mathbf{w}$ (varying the impact of each feature in fine-tuning). Thus, the network output is $f_\mathbf{w}(\mathbf{x}) = \sum_{j=1}^d \mathbf{w}_j \tilde{\mathbf{z}}_j(\mathbf{x})$. We consider two objective functions. The first is traditional ERM with weight decay ($\ell_2$ norm regularization). The ERM-$\ell_2$ objective function is

$$\min_\mathbf{w} \mathcal{L}^\lambda(\mathbf{w}) := \mathcal{L}(\mathbf{w}) + \frac{\lambda}{2}\|\mathbf{w}\|_2^2, \tag{3}$$

where $\mathcal{L}_{(\mathbf{x},y)}(\mathbf{w}) = \ell(y f_\mathbf{w}(\mathbf{x}))$ is the loss on an example $(\mathbf{x}, y)$ and $\ell(z)$ is the logistic loss $\ell(z) = \ln(1 + \exp(-z))$. The second objective we consider is ERM with bounded rank. Note that for a batch input data $\mathbf{X}$ with batch size $> d$, it is full rank with probability 1. Thus, we say the total feature rank is $\|\mathbf{w}\|_0 \leq d$ ($\|\mathbf{w}\|_0$ indicates the number of nonzero elements in $\mathbf{w}$). Thus, equivalent to Equation (1), with a upper bound $B_{\mathrm{rank}}$, the ERM-rank objective function is

$$\min_\mathbf{w} \mathcal{L}(\mathbf{w}) \quad \text{subject to} \quad \|\mathbf{w}\|_0 \leq B_{\mathrm{rank}}. \tag{4}$$

Note that our method, i.e., Equation (2), is a convex relaxation to the ERM-rank objective function.

**Theoretical results.** First, we analyze the property of the optimal solution of ERM-$\ell_2$ on the ID task. Following the idea from Lemma B.1 of [47], we have the Lemma below.

**Lemma 1.** *Consider the ID setting with the ERM-$\ell_2$ objective function. Then any optimal $\mathbf{w}^*$ of ERM-$\ell_2$ objective function follows conditions (1) for any $j \in R$, $\mathbf{w}_j^* =: \alpha$; (2) for any $j \in U$, $\mathbf{w}_j^* := \beta$; (3) $0 < \beta < \alpha < \frac{1}{\sqrt{r}}$, $\frac{\alpha}{\beta} < \frac{3}{4\gamma}$.*

In Lemma 1, we show that the ERM-$\ell_2$ objective will encode all features correlated with labels, even when the correlation between spurious features and labels is weak (e.g. $\gamma = O(1/\sqrt{r})$). However, the optimal solution of the ERM-rank objective will only encode the features that have a strong correlation with labels (invariant features), shown in Lemma 2.

**Lemma 2.** *Assume $1 \leq B_{\mathrm{rank}} \leq r$. Consider the ID setting with the ERM-rank objective function. For any optimal $\mathbf{w}^*$ of ERM-rank objective function, let $R_{\mathrm{rank}} = \{j \in [d] : \mathbf{w}_j^* \neq 0\}$. Then, we have $R_{\mathrm{rank}}$ satisfying the following property (1) $|R_{\mathrm{rank}}| = B_{\mathrm{rank}}$ and (2) $R_{\mathrm{rank}} \subseteq R$.*

Based on the property of two optimal solutions, we can show the performance gap between these two optimal solutions on the OOD task, considering the spurious features may change their correlation to the labels in different tasks.

**Proposition 3.** *Assume $1 \leq B_{\mathrm{rank}} \leq r, \lambda > \Omega\left(\frac{\sqrt{r}}{\exp\left(\frac{\sqrt{r}}{5}\right)}\right), d > \frac{r}{\gamma^2} + r, r > C$, where $C$ is some constant $< 20$. The optimal solution for the ERM-rank objective function on the ID tasks has 100% OOD test accuracy, while the optimal solution for the ERM-$\ell_2$ objective function on the ID tasks has OOD test accuracy at most $\exp\left(-\frac{r}{10}\right) \times 100\%$ (much worse than random guessing).*

**Discussions.** The assumption of $\lambda$ and $d$ means that the regularization strength cannot be too small and the environmental features signal level should be compatible with invariant features signal level. Then, Proposition 3 shows that even with infinite data, the optimal solution for ERM-$\ell_2$ on the ID tasks cannot produce better performance than random guessing on the OOD task. However, the optimal solution for ERM-rank on the ID tasks can still produce 100% test accuracy. The proof idea is that, by using the gradient equal to zero and the properties of the logistic loss, the ERM-$\ell_2$ objective will encode all features correlated with labels, even when the correlation between spurious features and label is weak (e.g. $\gamma = O\left(1/\sqrt{r}\right)$). Moreover, there is a positive correlation between the feature encoding strength and the corresponding feature-label correlation (Lemma 1 (3)). Then, we can show that the value of $\beta$ is compatible with the value of $\alpha$ in Lemma 1. Thus, when the OOD tasks have a different spurious feature distribution, the optimal solution of ERM-$\ell_2$ objective may thoroughly fail, i.e., much worse than random guessing. However, the optimal solution of the ERM-rank objective will only encode the features that have a strong correlation with labels (invariant features). Thus, it can guarantee 100% test accuracy on OOD tasks. See the full proof in Appendix D.

## 5. Related Works

**Nuclear norm minimization.** Nuclear norm is commonly used to approximate the matrix rank [22]. Nuclear norm minimization has been widely used in many areas where the solution is expected to have a low-rank structure. It has been widely applied for low-rank matrix approximation, completion, and recovery [16–19] with applications such as graph clustering [20], community detection [48], compressed sensing [49], recommendations system [50] and robust Principal Component Analysis [51]. Nuclear norm regularization can also be used in multi-task learning to learn shared representations across multiple tasks, which can lead to improved generalization and reduce overfitting [52]. Nuclear norm has been used in computer vision as well to solve problems such as image denoising [21] and image restoration [53]. In this work, we focus on utilizing nuclear norm-based regularization for domain generalization. We provide extensive experiments on synthetic and realistic datasets and theoretical analysis to better understand their effectiveness.

**Contextual bias, domain generalization, and group robustness.** There has been rich literature studying the classification performance in the presence of pre-defined contextual bias and spurious correlations [2, 45, 54–56]. The reliance on contextual bias such as image backgrounds, texture, and color for object detection has also been explored [28, 57–61]. In contrast, our study requires no prior information on the type of contextual bias and is broadly applicable to different categories of bias. The task of domain generalization aims to improve the classification performance of models on new test domains. A plethora of algorithms have been proposed in recent years: learning domain invariant [7–10, 62] and domain-specific features [39], minimizing the weighted combination of risks from training domains [28], mixing risk penalty terms to facilitate invariance prediction [6, 14], prototype-based contrastive learning [63], meta-learning [64], and data-centric approaches such as generation [65] and mixup [1, 3–5]. Recent works also demonstrate promising results with pre-trained models [40, 66–71]. Beyond domain generalization, another closely related task is to improve the group robustness in the presence of spurious correlations [28, 72, 73]. However, recent works often assume access to group labels for a small dataset or require multiple stages of training. In contrast, our approach is simple and efficient, requiring no access to domain labels or multi-stage training, and can improve over ERM-like algorithms on a broad range of real-world datasets.

## 6. Conclusions

In this work, we propose nuclear norm minimization, a simple yet effective regularization method for improving domain generalization without acquiring domain annotations. Key to our method is minimizing the nuclear norm of feature embeddings as a convex proxy for rank minimization. Empirically, our method is broadly applicable to many competitive algorithms for domain generalization and achieves competitive performance across synthetic and a wide range of real-world datasets. Theoretically, our method outperforms ERM with $\ell_2$ regularization in the linear setting.

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

# Appendix

## A. Acknowledgements

The work is partially supported by Air Force Grant FA9550-18-1-0166, the National Science Foundation (NSF) Grants 2008559-IIS, CCF-2046710, CCF-2106707. We are grateful for the support of the Wisconsin Alumni Research Foundation (WARF).

## B. Broader Impact

Our work aims at improving the domain generalization performance of models. Our paper is purely theoretical and empirical in nature and thus we foresee no immediate negative ethical impact. We provide a simple yet effective method that can be applied to different models, which may have a positive impact on the machine learning community. We hope our work will inspire effective algorithm design and promote a better understanding of domain generalization.

## C. Limitation

Our theoretical analysis requires strong assumptions on data distribution, e.g., coordinate independence and uniform distribution, although it is more general than the toy data model defined in Section 3.1. Our analysis cannot fully explain or apply to the model train on real-world datasets that contain non-linear data, e.g., DomainNet [46], but we are trying to provide some insights into why nuclear norm regularization can be more robust than the ERM solution by using a simple linear data model. On the other hand, studying the general necessary and sufficient condition of domain generalization is still an open challenging problem [74]. We believe it may be beyond the scope of this paper and we leave it as future work.

## D. Proof of Theoretical Analysis

### D.1. Auxiliary lemmas

We first present some Lemmas we will use later.

**Lemma 4.** *For the logistic loss $\ell(z) = \ln(1 + \exp(-z))$, we have the following statements (1) $\ell(z)$ is strictly decreasing and convex function on $\mathbb{R}$ and $\ell(z) > 0$; (2) $\ell'(z) = \frac{-1}{1+\exp(z)}$, $\ell'(z) \in (-1, 0)$; (3) $\ell'(z)$ is strictly concave on $[0, +\infty)$, (4) for any $c > 0$, $\ell'(z + c) \le \exp(-c)\ell'(z)$.*

*Proof of Lemma 4.* These can be verified by direct calculation. $\qquad\square$

**Lemma 5.**
$$\frac{\partial \mathcal{L}_{(\mathbf{x},y)}(\mathbf{w})}{\partial \mathbf{w}_j} = \ell'(y f_{\mathbf{w}}(\mathbf{x}))\mathbf{z}_j, \tag{5}$$

$$\frac{\partial \mathcal{L}(\mathbf{w})}{\partial \mathbf{w}_j} = \mathbb{E}_{(\mathbf{x},y)}\left[\ell'(y f_{\mathbf{w}}(\mathbf{x}))\mathbf{z}_j\right] \tag{6}$$

$$\frac{\partial \mathcal{L}^{\lambda}(\mathbf{w})}{\partial \mathbf{w}_j} = \mathbb{E}_{(\mathbf{x},y)}\left[\ell'(y f_{\mathbf{w}}(\mathbf{x}))\mathbf{z}_j\right] + \lambda \mathbf{w}_j \tag{7}$$

*Proof of Lemma 5.* These can be verified by direct calculation. $\qquad\square$

**Lemma 6.** *For any $j \in R$, we have probability density function of $\mathbf{z}_j$ with mean $\frac{1}{2}$ and variance $\frac{1}{12}$ following the form*

$$f_{\{\mathbf{z}_j\}}(z) = \begin{cases} 1, & if \quad 0 \le z \le 1 \\ 0, & otherwise . \end{cases}$$

For any $j \in U$, we have probability density function of $\mathbf{z}_j$ with mean $\gamma$ and variance $\frac{1}{3} - \gamma^2$ following the form

$$f_{\{\mathbf{z}_j\}}(z) = \begin{cases} \frac{1}{2} - \gamma, & if \quad -1 \leq z < 0 \\ \frac{1}{2} + \gamma, & if \quad 0 \leq z \leq 1 \\ 0, & otherwise . \end{cases}$$

*Proof of Lemma 6.* Then these can be verified by direct calculation from the definition. $\qquad\square$

**Lemma 7.** *We have* $\mathbb{P}\left[\sum_{j \in U} \mathbf{z}_j \leq 0\right] \leq \exp\left(-\frac{(d-r)\gamma^2}{2}\right)$, $\mathbb{P}\left[\sum_{j \in R} \mathbf{z}_j \leq \frac{r}{4}\right] \leq \exp\left(-\frac{r}{8}\right)$.

*Proof of Lemma 7.* By Hoeffding's inequality,

$$\mathbb{P}\left[\sum_{j \in U} \mathbf{z}_j \leq 0\right] = \mathbb{P}\left[\sum_{j \in U}(\mathbf{z}_j - \gamma) \leq -(d-r)\gamma\right] \tag{8}$$

$$\leq \exp\left(-\frac{(d-r)\gamma^2}{2}\right). \tag{9}$$

The others are proven in a similar way. $\qquad\square$

## D.2. Optimal solution of ERM-$\ell_2$ on ID task

**Lemma 8** (Restatement of Lemma 1 (1)(2)). *Consider the ID setting with the ERM-$\ell_2$ objective function. Then any optimal $\mathbf{w}^*$ of ERM-$\ell_2$ objective function follows conditions (1) for any $j \in R$, $\mathbf{w}_j^* =: \alpha$ and (2) for any $j \in U$, $\mathbf{w}_j^* := \beta$.*

*Proof of Lemma 8.*

$$\mathcal{L}^\lambda(\mathbf{w}^*) = \mathbb{E}_{(\mathbf{x},y)\sim\mathcal{D}_{\mathrm{id}}}\mathcal{L}_{(\mathbf{x},y)}(\mathbf{w}^*) + \frac{\lambda}{2}\|\mathbf{w}^*\|_2^2$$

$$= \mathbb{E}_{(\mathbf{x},y)\sim\mathcal{D}_{\mathrm{id}}}\ell(yf_{\mathbf{w}^*}(\mathbf{x})) + \frac{\lambda}{2}\|\mathbf{w}^*\|_2^2$$

$$= \mathbb{E}_{(\mathbf{x},y)\sim\mathcal{D}_{\mathrm{id}}}\ell\left(\sum_{j=1}^d \mathbf{w}_j^*\mathbf{z}_j\right) + \frac{\lambda}{2}\|\mathbf{w}^*\|_2^2$$

By Lemma 4, we have $\mathcal{L}^\lambda(\mathbf{w})$ a is convex function. By symmetry of $\mathbf{z}_j$, for any $l, l' \in R, l \neq l'$,

$$\mathbb{E}\left[\ell\left(\sum_{j=1}^d \mathbf{w}_j^*\mathbf{z}_j\right)\right] + \frac{\lambda}{2}\|\mathbf{w}^*\|_2^2 \tag{10}$$

$$= \frac{1}{2}\left(\mathbb{E}\left[\ell\left(\sum_{j\in[d],j\neq l,j\neq l'} \mathbf{w}_j^*\mathbf{z}_j + \mathbf{w}_l^*\mathbf{z}_l + \mathbf{w}_{l'}^*\mathbf{z}_{l'}\right)\right] + \frac{\lambda}{2}\|\mathbf{w}^*\|_2^2\right) \tag{11}$$

$$+ \frac{1}{2}\left(\mathbb{E}\left[\ell\left(\sum_{j\in[d],j\neq l,j\neq l'} \mathbf{w}_j^*\mathbf{z}_j + \mathbf{w}_l^*\mathbf{z}_{l'} + \mathbf{w}_{l'}^*\mathbf{z}_l\right)\right] + \frac{\lambda}{2}\|\mathbf{w}^*\|_2^2\right) \tag{12}$$

$$\geq \mathbb{E}\left[\ell\left(\sum_{j\in[d],j\neq l,j\neq l'} \mathbf{w}_j^*\mathbf{z}_j + \frac{\mathbf{w}_l^* + \mathbf{w}_{l'}^*}{2}\mathbf{z}_{l'} + \frac{\mathbf{w}_l^* + \mathbf{w}_{l'}^*}{2}\mathbf{z}_l\right)\right] + \frac{\lambda}{2}\|\mathbf{w}^*\|_2^2, \tag{13}$$

where the last inequality follows Jensen's inequality. Note that the last equation is true only when $\mathbf{z}_l$ and $\mathbf{z}_{l'}$ share the same distribution. The minimum is achieved when $\mathbf{w}_l^* = \mathbf{w}_{l'}^*$.

A similar argument as above proves statement (2). $\qquad\square$

Now, we will bound the $\alpha$ and $\beta$. Recall that for any $j \in R$, $\mathbf{w}_j^* =: \alpha$ and for any $j \in U$, $\mathbf{w}_j^* := \beta$.

**Lemma 9** (Restatement of Lemma 1 (3)). *Let $\alpha, \beta$ be values defined in the Lemma 8. Then, we have $0 < \beta < \alpha < \frac{1}{\sqrt{r}}$. Moreover, $\frac{\alpha}{\beta} < \frac{3}{4\gamma}$.*

*Proof of Lemma 9.* By Lemma 8

$$\mathcal{L}^\lambda(\mathbf{w}^*) = \mathbb{E}\left[\ell\left(\alpha \sum_{j \in R} \mathbf{z}_j + \beta \sum_{j \in U} \mathbf{z}_j\right)\right] + \frac{\lambda}{2}(r\alpha^2 + (d - r)\beta^2) \tag{14}$$

$$= \mathcal{L}^\lambda(\alpha, \beta). \tag{15}$$

By Lemma 5, we have for any $j \in [d]$

$$\frac{\partial \mathcal{L}^\lambda(\mathbf{w}^*)}{\partial \mathbf{w}_j^*} = \mathbb{E}_{(\mathbf{x}, y) \sim \mathcal{D}_{\mathrm{id}}}\left[\ell'(y f_\mathbf{w}^*(\mathbf{x}))\mathbf{z}_j\right] + \lambda \mathbf{w}_j^* = 0. \tag{16}$$

We first prove $\beta < \alpha$. For any $j \in R, j' \in U$, we have

$$\lambda\alpha = \lambda\mathbf{w}_j^* \tag{17}$$

$$= -\mathbb{E}_{(\mathbf{x}, y) \sim \mathcal{D}_{\mathrm{id}}}\left[\ell'(y f_\mathbf{w}^*(\mathbf{x}))\mathbf{z}_j\right] \tag{18}$$

$$> -\mathbb{E}_{(\mathbf{x}, y) \sim \mathcal{D}_{\mathrm{id}}}\left[\ell'(y f_\mathbf{w}^*(\mathbf{x}))\mathbf{z}_{j'}(\mathbf{x}, y)\right] \tag{19}$$

$$= \lambda\mathbf{w}_{j'}^* = \lambda\beta. \tag{20}$$

Then, we prove $\beta \geq 0$ by contradiction. Suppose $\beta < 0$,

$$\mathcal{L}^\lambda(\alpha, \beta) - \mathcal{L}^\lambda(\alpha, -\beta) \tag{21}$$

$$= \mathbb{E}\left[\ell\left(\alpha \sum_{j \in R} \mathbf{z}_j + \beta \sum_{j \in U} \mathbf{z}_j\right)\right] - \mathbb{E}\left[\ell\left(\alpha \sum_{j \in R} \mathbf{z}_j - \beta \sum_{j \in U} \mathbf{z}_j\right)\right]. \tag{22}$$

Note that for any $j, j' \in U, j \neq j'$, the norm of $\mathbf{z}_j$ is independent with its sign and $\mathbf{z}_j, \mathbf{z}_{j'}$ are independent. From $\gamma > 0$, we can get $\mathbb{P}[\mathbf{z}_j > 0] > \frac{1}{2}$. Thus, by $\ell$ strictly decreasing we have

$$\mathbb{P}\left[\ell\left(\alpha \sum_{j \in R} \mathbf{z}_j + \beta \sum_{j \in U} \mathbf{z}_j\right) \geq z\right] > \mathbb{P}\left[\ell\left(\alpha \sum_{j \in R} \mathbf{z}_j - \beta \sum_{j \in U} \mathbf{z}_j\right) \geq z\right], \tag{23}$$

where $\beta$ case is strictly stochastically dominate $-\beta$ case. Thus, $\mathcal{L}^\lambda(\alpha, \beta) - \mathcal{L}^\lambda(\alpha, -\beta) > 0$. This is contradicted by $\beta$ being the optimal value. Thus, we have $\beta \geq 0$.

Now, we prove $\alpha < \frac{1}{\sqrt{r}}$, for any $k \in R$,

$$\lambda\alpha = -\mathbb{E}_{(\mathbf{x},y)\sim\mathcal{D}_{\text{id}}}\left[\ell'\left(\alpha\sum_{j\in R}\mathbf{z}_j + \beta\sum_{j\in U}\mathbf{z}_j\right)\mathbf{z}_k\right] \tag{24}$$

$$\leq -\mathbb{E}_{(\mathbf{x},y)\sim\mathcal{D}_{\text{id}}}\left[\ell'\left(\alpha\sum_{j\in R,j\neq k}\mathbf{z}_j + \beta\sum_{j\in U}\mathbf{z}_j\right)\mathbf{z}_k\right] \tag{25}$$

$$= -\mathbb{E}\left[\ell'\left(\alpha\sum_{j\in R,j\neq k}\mathbf{z}_j + \beta\sum_{j\in U}\mathbf{z}_j\right)\right]\mathbb{E}[\mathbf{z}_k] \tag{26}$$

$$= -\frac{1}{2}\mathbb{E}\left[\ell'\left(\alpha\sum_{j\in R,j\neq k}\mathbf{z}_j + \beta\sum_{j\in U}\mathbf{z}_j\right)\Bigg|\sum_{j\in U}\mathbf{z}_j > 0\right]\mathbb{P}\left[\sum_{j\in U}\mathbf{z}_j > 0\right] \tag{27}$$

$$\quad -\frac{1}{2}\mathbb{E}\left[\ell'\left(\alpha\sum_{j\in R,j\neq k}\mathbf{z}_j + \beta\sum_{j\in U}\mathbf{z}_j\right)\Bigg|\sum_{j\in U}\mathbf{z}_j \leq 0\right]\mathbb{P}\left[\sum_{j\in U}\mathbf{z}_j \leq 0\right] \tag{28}$$

$$\leq -\frac{1}{2}\mathbb{E}\left[\ell'\left(\alpha\sum_{j\in R,j\neq k}\mathbf{z}_j\right)\right] + \frac{1}{2}\exp\left(-\frac{(d-r)\gamma^2}{2}\right), \tag{29}$$

where the last inequality is from $\beta \geq 0$ and $\ell'(z) \in (-1, 0)$. Using Lemma 7 one more time, we have

$$-\mathbb{E}\left[\ell'\left(\alpha\sum_{j\in R,j\neq k}\mathbf{z}_j\right)\right] \tag{30}$$

$$= -\mathbb{E}\left[\ell'\left(\alpha\sum_{j\in R,j\neq k}\mathbf{z}_j\Bigg|\sum_{j\in R,j\neq k}\mathbf{z}_j > \frac{r-1}{4}\right)\right]\mathbb{P}\left[\sum_{j\in R,j\neq k}\mathbf{z}_j > \frac{r-1}{4}\right] \tag{31}$$

$$\quad -\mathbb{E}\left[\ell'\left(\alpha\sum_{j\in R,j\neq k}\mathbf{z}_j\Bigg|\sum_{j\in R,j\neq k}\mathbf{z}_j \leq \frac{r-1}{4}\right)\right]\mathbb{P}\left[\sum_{j\in R,j\neq k}\mathbf{z}_j \leq \frac{r-1}{4}\right] \tag{32}$$

$$\leq -\ell'\left(\frac{\alpha(r-1)}{4}\right) + \frac{1}{2}\exp\left(-\frac{r-1}{8}\right) \tag{33}$$

$$= \frac{1}{1+\exp\left(\frac{\alpha(r-1)}{4}\right)} + \frac{1}{2}\exp\left(-\frac{r-1}{8}\right). \tag{34}$$

Thus, we have

$$\lambda\alpha \leq \frac{1}{2\left(1+\exp\left(\frac{\alpha(r-1)}{4}\right)\right)} + \frac{1}{4}\exp\left(-\frac{r-1}{8}\right) + \frac{1}{2}\exp\left(-\frac{(d-r)\gamma^2}{2}\right). \tag{35}$$

Suppose $\alpha \geq \frac{1}{\sqrt{r}}$, we have contradiction,

$$\text{RHS} < O\left(\exp\left(-\frac{\sqrt{r}}{5}\right)\right) < \text{LHS}. \tag{36}$$

Thus, we get $\alpha < \frac{1}{\sqrt{r}}$.

Now, we prove $\frac{\alpha}{\beta} \leq \frac{3}{4\gamma}$, for any $k \in R, l \in U$, denote $Z = \alpha \sum_{j \in R, j \neq k} \mathbf{z}_j + \beta \sum_{j \in U, j \neq l} \mathbf{z}_j$, by Lemma 4, we have

$$
\frac{\alpha}{\beta} = \frac{-\mathbb{E}\left[\ell'\left(\alpha \sum_{j \in R} \mathbf{z}_j + \beta \sum_{j \in U} \mathbf{z}_j\right) \mathbf{z}_k\right]}{-\mathbb{E}\left[\ell'\left(\alpha \sum_{j \in R} \mathbf{z}_j + \beta \sum_{j \in U} \mathbf{z}_j\right) \mathbf{z}_l\right]} \tag{37}
$$

$$
\leq \frac{-\mathbb{E}\left[\ell'\left(Z\right) \mathbf{z}_k\right]}{-\mathbb{E}\left[\ell'\left(Z + 2\alpha\right) \mathbf{z}_l | \mathbf{z}_l \geq 0\right] \mathbb{P}[\mathbf{z}_l \geq 0] - \mathbb{E}\left[\ell'\left(Z\right) \mathbf{z}_l | \mathbf{z}_l < 0\right] \mathbb{P}[\mathbf{z}_l < 0]} \tag{38}
$$

$$
= \frac{-\mathbb{E}\left[\ell'\left(Z\right)\right]}{-\mathbb{E}\left[\ell'\left(Z + 2\alpha\right)\right]\left(\frac{1}{2} + \gamma\right) + \mathbb{E}\left[\ell'\left(Z\right)\right]\left(\frac{1}{2} - \gamma\right)} \tag{39}
$$

$$
\leq \frac{-\mathbb{E}\left[\ell'\left(Z\right)\right]}{-\exp(-2\alpha)\mathbb{E}\left[\ell'\left(Z\right)\right]\left(\frac{1}{2} + \gamma\right) + \mathbb{E}\left[\ell'\left(Z\right)\right]\left(\frac{1}{2} - \gamma\right)} \tag{40}
$$

$$
= \frac{1}{\exp(-2\alpha)\left(\frac{1}{2} + \gamma\right) - \left(\frac{1}{2} - \gamma\right)} \tag{41}
$$

$$
\leq \frac{1}{\exp\left(\frac{-2}{\sqrt{r}}\right)\left(\frac{1}{2} + \gamma\right) - \left(\frac{1}{2} - \gamma\right)} \tag{42}
$$

$$
\leq \frac{1}{2\gamma - \left(1 - \exp\left(\frac{-2}{\sqrt{r}}\right)\right)} \tag{43}
$$

$$
\leq \frac{1}{2\gamma - \frac{2}{\sqrt{r}}} \tag{44}
$$

$$
< \frac{3}{4\gamma}, \tag{45}
$$

where the second inequality follows Lemma 4 and the second last inequality follows $1 + z \leq \exp(z)$ for $z \in \mathbb{R}$ and $\gamma > \frac{3}{\sqrt{r}}$. $\qquad \square$

## D.3. Optimal solution of ERM-rank on ID task

**Lemma 10** (Restatement of Lemma 2). *Assume $1 \leq B_{\text{rank}} \leq r$. Consider the ID setting with the ERM-rank objective function. For any optimal $\mathbf{w}^*$ of ERM-rank objective function, let $R_{\text{rank}} = \{j \in [d] : \mathbf{w}_j^* \neq 0\}$. Then, we have $R_{\text{rank}}$ satisfying the following property (1) $|R_{\text{rank}}| = B_{\text{rank}}$ and (2) $R_{\text{rank}} \subseteq R$.*

*Proof of Lemma 10.* For any $j \in U$, if $\mathbf{w}_j^* = \theta \neq 0$, there exists $k \in R$ s.t. $\mathbf{w}_k^* = 0$ by objective function condition. When we reassign $\mathbf{w}_j^* = 0, \mathbf{w}_k^* = |\theta|$, the objective function becomes smaller. This is a contradiction. Thus, we finish the proof. $\qquad \square$

## D.4. OOD gap between two objective function

**Proposition 11** (Restatement of Proposition 3). *Assume $1 \leq B_{\text{rank}} \leq r, \lambda > \Omega\left(\frac{\sqrt{r}}{\exp\left(\frac{\sqrt{r}}{5}\right)}\right), d > \frac{r}{\gamma^2} + r, r > C$, where $C$ is some constant $< 20$. The optimal solution for the ERM-rank objective function on the ID tasks has 100% OOD test accuracy, while the optimal solution for the ERM-$\ell_2$ objective function on the ID tasks has OOD test accuracy at most $\exp\left(-\frac{r}{10}\right) \times 100\%$ (much worse than random guessing).*

*Proof of Proposition 11.* We denote $\mathbf{w}^*_{rank}$ as the optimal solution for the ERM-rank objective function. By Lemma 10, the test accuracy for the ERM-rank objective function is

$$\mathbb{P}_{(\mathbf{x},y)\sim\mathcal{D}_{\text{ood}}}[yf_{\mathbf{w}^*_{rank}}(\mathbf{x})\geq 0] =\mathbb{P}_{(\mathbf{x},y)\sim\mathcal{D}_{\text{ood}}}\left[\sum_{j\in R}\mathbf{w}^*_{rank,j}\mathbf{z}_j + \sum_{j\in U}\mathbf{z}_j\mathbf{w}^*_{rank,j}\geq 0\right] \tag{46}$$

$$=\mathbb{P}_{(\mathbf{x},y)\sim\mathcal{D}_{\text{ood}}}\left[\sum_{j\in R}\mathbf{w}^*_{rank,j}\mathbf{z}_j \geq 0\right] \tag{47}$$

$$=1. \tag{48}$$

We denote $\mathbf{w}^*_{\ell_2}$ as the optimal solution for the ERM-rank objective function. We have $\alpha, \beta$ defined in Lemma 9. By Lemma 9, the test accuracy for the ERM-$\ell_2$ objective function is

$$\mathbb{P}_{(\mathbf{x},y)\sim\mathcal{D}_{\text{ood}}}[yf_{\mathbf{w}^*_{\ell_2}}(\mathbf{x})\geq 0] =\mathbb{P}_{(\mathbf{x},y)\sim\mathcal{D}_{\text{ood}}}\left[\alpha\sum_{j\in R}\mathbf{z}_j + \beta\sum_{j\in U}\mathbf{z}_j \geq 0\right] \tag{49}$$

$$\leq\mathbb{P}_{(\mathbf{x},y)\sim\mathcal{D}_{\text{ood}}}\left[\frac{3}{4\gamma}\sum_{j\in R}\mathbf{z}_j + \sum_{j\in U}\mathbf{z}_j \geq 0\right] \tag{50}$$

$$=\mathbb{P}\left[\frac{3}{4\gamma}\sum_{j\in R}\left(\mathbf{z}_j - \frac{1}{2}\right) + \sum_{j\in U}(\mathbf{z}_j + \gamma) \geq -\frac{3r}{8\gamma} + (d-r)\gamma\right] \tag{51}$$

By Hoeffding's inequality and $d > \frac{r}{\gamma^2} + r > 5r$, we have

$$\mathbb{P}\left[\frac{3}{4\gamma}\sum_{j\in R}\left(\mathbf{z}_j - \frac{1}{2}\right) + \sum_{j\in U}(\mathbf{z}_j + \gamma) \geq -\frac{3r}{8\gamma} + (d-r)\gamma\right] \tag{52}$$

$$\leq\exp\left(-\frac{2\left(-\frac{3r}{8\gamma} + (d-r)\gamma\right)^2}{4d}\right) \tag{53}$$

$$=\exp\left(-\frac{\frac{9r^2}{32\gamma^2} + 2(d-r)^2\gamma^2 - \frac{3r}{2}(d-r)}{4d}\right) \tag{54}$$

$$\leq\exp\left(-\frac{2(d-r)^2\gamma^2 - \frac{3r}{2}(d-r)}{5(d-r)}\right) \tag{55}$$

$$=\exp\left(-\frac{4(d-r)\gamma^2 - 3r}{10}\right) \tag{56}$$

$$\leq\exp\left(-\frac{r}{10}\right). \tag{57}$$

$\square$

# E. More Experiments Details and Results

| Algorithm | VLCS | PACS | OfficeHome | TerraInc | DomainNet | Average |
|---|---|---|---|---|---|---|
| SWAD | $79.1 \pm 0.1$ | $88.1 \pm 0.1$ | $70.6 \pm 0.2$ | $50.0 \pm 0.3$ | $46.5 \pm 0.1$ | 66.9 |
| SWAD-CORAL | $78.9 \pm 0.1$ | $88.3 \pm 0.1$ | $\underline{71.3} \pm 0.1$ | $51.0 \pm 0.1$ | $46.8 \pm 0.0$ | 67.3 |
| SWAD-MIRO | $\underline{79.6} \pm 0.2$ | $\underline{88.4} \pm 0.1$ | $\mathbf{72.4} \pm 0.1$ | $\mathbf{52.9} \pm 0.2$ | $\underline{47.0} \pm 0.0$ | $\mathbf{68.1}$ |
| SWAD-NU (ours) | $\mathbf{79.8} \pm 0.2$ | $\mathbf{88.5} \pm 0.2$ | $\underline{71.3} \pm 0.3$ | $\underline{52.2} \pm 0.3$ | $\mathbf{47.1} \pm 0.1$ | $\underline{67.8}$ |

Table 4: Methods combined with SWAD full results on DomainBed benchmark.

| Algorithm | C | L | S | V | Average |
|---|---|---|---|---|---|
| IRM | $98.6 \pm 0.1$ | $64.9 \pm 0.9$ | $73.4 \pm 0.6$ | $77.3 \pm 0.9$ | 78.5 |
| GroupDRO | $97.3 \pm 0.3$ | $63.4 \pm 0.9$ | $69.5 \pm 0.8$ | $76.7 \pm 0.7$ | 76.7 |
| MLDG | $97.4 \pm 0.2$ | $65.2 \pm 0.7$ | $71.0 \pm 1.4$ | $75.3 \pm 1.0$ | 77.2 |
| CORAL | $98.3 \pm 0.1$ | $\underline{66.1} \pm 1.2$ | $73.4 \pm 0.3$ | $77.5 \pm 1.2$ | 78.8 |
| MMD | $97.7 \pm 0.1$ | $64.0 \pm 1.1$ | $72.8 \pm 0.2$ | $75.3 \pm 3.3$ | 77.5 |
| DANN | $\underline{99.0} \pm 0.3$ | $65.1 \pm 1.4$ | $73.1 \pm 0.3$ | $77.2 \pm 0.6$ | 78.6 |
| CDANN | $97.1 \pm 0.3$ | $65.1 \pm 1.2$ | $70.7 \pm 0.8$ | $77.1 \pm 1.5$ | 77.5 |
| MTL | $97.8 \pm 0.4$ | $64.3 \pm 0.3$ | $71.5 \pm 0.7$ | $75.3 \pm 1.7$ | 77.2 |
| SagNet | $97.9 \pm 0.4$ | $64.5 \pm 0.5$ | $71.4 \pm 1.3$ | $77.5 \pm 0.5$ | 77.8 |
| ARM | $98.7 \pm 0.2$ | $63.6 \pm 0.7$ | $71.3 \pm 1.2$ | $76.7 \pm 0.6$ | 77.6 |
| VREx | $98.4 \pm 0.3$ | $64.4 \pm 1.4$ | $74.1 \pm 0.4$ | $76.2 \pm 1.3$ | 78.3 |
| RSC | $97.9 \pm 0.1$ | $62.5 \pm 0.7$ | $72.3 \pm 1.2$ | $75.6 \pm 0.8$ | 77.1 |
| AND-mask | $97.8 \pm 0.4$ | $64.3 \pm 1.2$ | $73.5 \pm 0.7$ | $76.8 \pm 2.6$ | 78.1 |
| SelfReg | $96.7 \pm 0.4$ | $65.2 \pm 1.2$ | $73.1 \pm 1.3$ | $76.2 \pm 0.7$ | 77.8 |
| mDSDI | $97.6 \pm 0.1$ | $\mathbf{66.4} \pm 0.4$ | $74.0 \pm 0.6$ | $77.8 \pm 0.7$ | 79.0 |
| Fishr | $98.9 \pm 0.3$ | $64.0 \pm 0.5$ | $71.5 \pm 0.2$ | $76.8 \pm 0.7$ | 77.8 |
| ERM | $97.7 \pm 0.4$ | $64.3 \pm 0.9$ | $73.4 \pm 0.5$ | $74.6 \pm 1.3$ | 77.5 |
| ERM-NU (ours) | $97.9 \pm 0.4$ | $65.1 \pm 0.3$ | $73.2 \pm 0.9$ | $76.9 \pm 0.5$ | 78.3 |
| Mixup | $98.3 \pm 0.6$ | $64.8 \pm 1.0$ | $72.1 \pm 0.5$ | $74.3 \pm 0.8$ | 77.4 |
| Mixup-NU (ours) | $97.9 \pm 0.2$ | $64.1 \pm 1.4$ | $73.1 \pm 0.9$ | $74.8 \pm 0.5$ | 77.5 |
| SWAD | $98.8 \pm 0.1$ | $63.3 \pm 0.3$ | $\underline{75.3} \pm 0.5$ | $\underline{79.2} \pm 0.6$ | $\underline{79.1}$ |
| SWAD-NU (ours) | $\mathbf{99.1} \pm 0.4$ | $63.6 \pm 0.4$ | $\mathbf{75.9} \pm 0.4$ | $\mathbf{80.5} \pm 1.0$ | $\mathbf{79.8}$ |

Table 5: Results on VLCS. For each column, bold indicates the best performance, and underline indicates the second-best performance.

| Algorithm | A | C | P | S | Average |
|---|---|---|---|---|---|
| IRM | 84.8 ± 1.3 | 76.4 ± 1.1 | 96.7 ± 0.6 | 76.1 ± 1.0 | 83.5 |
| GroupDRO | 83.5 ± 0.9 | 79.1 ± 0.6 | 96.7 ± 0.3 | 78.3 ± 2.0 | 84.4 |
| MLDG | 85.5 ± 1.4 | 80.1 ± 1.7 | 97.4 ± 0.3 | 76.6 ± 1.1 | 84.9 |
| CORAL | 88.3 ± 0.2 | 80.0 ± 0.5 | 97.5 ± 0.3 | 78.8 ± 1.3 | 86.2 |
| MMD | 86.1 ± 1.4 | 79.4 ± 0.9 | 96.6 ± 0.2 | 76.5 ± 0.5 | 84.6 |
| DANN | 86.4 ± 0.8 | 77.4 ± 0.8 | 97.3 ± 0.4 | 73.5 ± 2.3 | 83.6 |
| CDANN | 84.6 ± 1.8 | 75.5 ± 0.9 | 96.8 ± 0.3 | 73.5 ± 0.6 | 82.6 |
| MTL | 87.5 ± 0.8 | 77.1 ± 0.5 | 96.4 ± 0.8 | 77.3 ± 1.8 | 84.6 |
| SagNet | 87.4 ± 1.0 | 80.7 ± 0.6 | 97.1 ± 0.1 | 80.0 ± 0.4 | 86.3 |
| ARM | 86.8 ± 0.6 | 76.8 ± 0.5 | 97.4 ± 0.3 | 79.3 ± 1.2 | 85.1 |
| VREx | 86.0 ± 1.6 | 79.1 ± 0.6 | 96.9 ± 0.5 | 77.7 ± 1.7 | 84.9 |
| RSC | 85.4 ± 0.8 | 79.7 ± 1.8 | 97.6 ± 0.3 | 78.2 ± 1.2 | 85.2 |
| AND-mask | 85.3 ± 1.4 | 79.2 ± 2.0 | 96.9 ± 0.4 | 76.2 ± 1.4 | 84.4 |
| SelfReg | 87.9 ± 1.0 | 79.4 ± 1.4 | 96.8 ± 0.7 | 78.3 ± 1.2 | 85.6 |
| mDSDI | 87.7 ± 0.4 | 80.4 ± 0.7 | **98.1** ± 0.3 | 78.4 ± 1.2 | 86.2 |
| Fishr | 88.4 ± 0.2 | 78.7 ± 0.7 | 97.0 ± 0.1 | 77.8 ± 2.0 | 85.5 |
| ERM | 84.7 ± 0.4 | 80.8 ± 0.6 | 97.2 ± 0.3 | 79.3 ± 1.0 | 85.5 |
| ERM-NU (ours) | 87.4 ± 0.5 | 79.6 ± 0.9 | 96.3 ± 0.7 | 79.0 ± 0.5 | 85.6 |
| Mixup | 86.1 ± 0.5 | 78.9 ± 0.8 | 97.6 ± 0.1 | 75.8 ± 1.8 | 84.6 |
| Mixup-NU (ours) | 86.7 ± 0.3 | 78.0 ± 1.3 | 97.3 ± 0.3 | 77.3 ± 2.0 | 84.8 |
| SWAD | 89.3 ± 0.2 | **83.4** ± 0.6 | 97.3 ± 0.3 | 82.5 ± 0.5 | 88.1 |
| SWAD-NU (ours) | **89.8** ± 1.1 | 82.8 ± 1.0 | 97.7 ± 0.3 | **83.7** ± 1.1 | **88.5** |

Table 6: Results on PACS.

| Algorithm | A | C | P | R | Average |
|---|---|---|---|---|---|
| IRM | 58.9 ± 2.3 | 52.2 ± 1.6 | 72.1 ± 2.9 | 74.0 ± 2.5 | 64.3 |
| GroupDRO | 60.4 ± 0.7 | 52.7 ± 1.0 | 75.0 ± 0.7 | 76.0 ± 0.7 | 66.0 |
| MLDG | 61.5 ± 0.9 | 53.2 ± 0.6 | 75.0 ± 1.2 | 77.5 ± 0.4 | 66.8 |
| CORAL | 65.3 ± 0.4 | 54.4 ± 0.5 | 76.5 ± 0.1 | 78.4 ± 0.5 | 68.7 |
| MMD | 60.4 ± 0.2 | 53.3 ± 0.3 | 74.3 ± 0.1 | 77.4 ± 0.6 | 66.3 |
| DANN | 59.9 ± 1.3 | 53.0 ± 0.3 | 73.6 ± 0.7 | 76.9 ± 0.5 | 65.9 |
| CDANN | 61.5 ± 1.4 | 50.4 ± 2.4 | 74.4 ± 0.9 | 76.6 ± 0.8 | 65.8 |
| MTL | 61.5 ± 0.7 | 52.4 ± 0.6 | 74.9 ± 0.4 | 76.8 ± 0.4 | 66.4 |
| SagNet | 63.4 ± 0.2 | 54.8 ± 0.4 | 75.8 ± 0.4 | 78.3 ± 0.3 | 68.1 |
| ARM | 58.9 ± 0.8 | 51.0 ± 0.5 | 74.1 ± 0.1 | 75.2 ± 0.3 | 64.8 |
| VREx | 60.7 ± 0.9 | 53.0 ± 0.9 | 75.3 ± 0.1 | 76.6 ± 0.5 | 66.4 |
| RSC | 60.7 ± 1.4 | 51.4 ± 0.3 | 74.8 ± 1.1 | 75.1 ± 1.3 | 65.5 |
| AND-mask | 59.5 ± 1.1 | 51.7 ± 0.2 | 73.9 ± 0.4 | 77.1 ± 0.2 | 65.6 |
| SelfReg | 63.6 ± 1.4 | 53.1 ± 1.0 | 76.9 ± 0.4 | 78.1 ± 0.4 | 67.9 |
| mDSDI | 62.4 ± 0.5 | 54.4 ± 0.4 | 76.2 ± 0.5 | 78.3 ± 0.1 | 67.8 |
| Fishr | **68.1** ± 0.3 | 52.1 ± 0.4 | 76.0 ± 0.2 | 80.4 ± 0.2 | 69.2 |
| ERM | 61.3 ± 0.7 | 52.4 ± 0.3 | 75.8 ± 0.1 | 76.6 ± 0.3 | 66.5 |
| ERM-NU (ours) | 63.3 ± 0.2 | 54.2 ± 0.3 | 76.7 ± 0.2 | 78.2 ± 0.3 | 68.1 |
| Mixup | 62.4 ± 0.8 | 54.8 ± 0.6 | 76.9 ± 0.3 | 78.3 ± 0.2 | 68.1 |
| Mixup-NU (ours) | 64.3 ± 0.5 | 55.9 ± 0.6 | 76.9 ± 0.4 | 78.0 ± 0.6 | 68.8 |
| SWAD | 66.1 ± 0.4 | 57.7 ± 0.4 | 78.4 ± 0.1 | 80.2 ± 0.2 | 70.6 |
| SWAD-NU (ours) | 67.5 ± 0.3 | **58.4** ± 0.6 | **78.6** ± 0.9 | **80.7** ± 0.1 | **71.3** |

Table 7: Results on OfficeHome.

| Algorithm | L100 | L38 | L43 | L46 | Average |
|---|---|---|---|---|---|
| IRM | 54.6 ± 1.3 | 39.8 ± 1.9 | 56.2 ± 1.8 | 39.6 ± 0.8 | 47.6 |
| GroupDRO | 41.2 ± 0.7 | 38.6 ± 2.1 | 56.7 ± 0.9 | 36.4 ± 2.1 | 43.2 |
| MLDG | 54.2 ± 3.0 | 44.3 ± 1.1 | 55.6 ± 0.3 | 36.9 ± 2.2 | 47.7 |
| CORAL | 51.6 ± 2.4 | 42.2 ± 1.0 | 57.0 ± 1.0 | 39.8 ± 2.9 | 47.6 |
| MMD | 41.9 ± 3.0 | 34.8 ± 1.0 | 57.0 ± 1.9 | 35.2 ± 1.8 | 42.2 |
| DANN | 51.1 ± 3.5 | 40.6 ± 0.6 | 57.4 ± 0.5 | 37.7 ± 1.8 | 46.7 |
| CDANN | 47.0 ± 1.9 | 41.3 ± 4.8 | 54.9 ± 1.7 | 39.8 ± 2.3 | 45.8 |
| MTL | 49.3 ± 1.2 | 39.6 ± 6.3 | 55.6 ± 1.1 | 37.8 ± 0.8 | 45.6 |
| SagNet | 53.0 ± 2.9 | 43.0 ± 2.5 | 57.9 ± 0.6 | 40.4 ± 1.3 | 48.6 |
| ARM | 49.3 ± 0.7 | 38.3 ± 2.4 | 55.8 ± 0.8 | 38.7 ± 1.3 | 45.5 |
| VREx | 48.2 ± 4.3 | 41.7 ± 1.3 | 56.8 ± 0.8 | 38.7 ± 3.1 | 46.4 |
| RSC | 50.2 ± 2.2 | 39.2 ± 1.4 | 56.3 ± 1.4 | 40.8 ± 0.6 | 46.6 |
| AND-mask | 50.0 ± 2.9 | 40.2 ± 0.8 | 53.3 ± 0.7 | 34.8 ± 1.9 | 44.6 |
| SelfReg | 48.8 ± 0.9 | 41.3 ± 1.8 | 57.3 ± 0.7 | 40.6 ± 0.9 | 47.0 |
| mDSDI | 53.2 ± 3.0 | 43.3 ± 1.0 | 56.7 ± 0.5 | 39.2 ± 1.3 | 48.1 |
| Fishr | 50.2 ± 3.9 | 43.9 ± 0.8 | 55.7 ± 2.2 | 39.8 ± 1.0 | 47.4 |
| ERM | 49.8 ± 4.4 | 42.1 ± 1.4 | 56.9 ± 1.8 | 35.7 ± 3.9 | 46.1 |
| ERM-NU (ours) | 52.5 ± 1.2 | 45.0 ± 0.5 | 60.2 ± 0.2 | 40.7 ± 1.0 | 49.6 |
| Mixup | **59.6** ± 2.0 | 42.2 ± 1.4 | 55.9 ± 0.8 | 33.9 ± 1.4 | 47.9 |
| Mixup-NU (ours) | 55.1 ± 3.1 | 45.8 ± 0.7 | 56.4 ± 1.2 | 41.1 ± 0.6 | 49.6 |
| SWAD | 55.4 ± 0.0 | 44.9 ± 1.1 | 59.7 ± 0.4 | 39.9 ± 0.2 | 50.0 |
| SWAD-NU (ours) | 58.1 ± 3.3 | **47.7** ± 1.6 | **60.5** ± 0.8 | **42.3** ± 0.9 | **52.2** |

Table 8: Results on Terra Incognita.

| Algorithm | clip | info | paint | quick | real | sketch | Average |
|---|---|---|---|---|---|---|---|
| IRM | 48.5 ± 2.8 | 15.0 ± 1.5 | 38.3 ± 4.3 | 10.9 ± 0.5 | 48.2 ± 5.2 | 42.3 ± 3.1 | 33.9 |
| GroupDRO | 47.2 ± 0.5 | 17.5 ± 0.4 | 33.8 ± 0.5 | 9.3 ± 0.3 | 51.6 ± 0.4 | 40.1 ± 0.6 | 33.3 |
| MLDG | 59.1 ± 0.2 | 19.1 ± 0.3 | 45.8 ± 0.7 | 13.4 ± 0.3 | 59.6 ± 0.2 | 50.2 ± 0.4 | 41.2 |
| CORAL | 59.2 ± 0.1 | 19.7 ± 0.2 | 46.6 ± 0.3 | 13.4 ± 0.4 | 59.8 ± 0.2 | 50.1 ± 0.6 | 41.5 |
| MMD | 32.1 ± 13.3 | 11.0 ± 4.6 | 26.8 ± 11.3 | 8.7 ± 2.1 | 32.7 ± 13.8 | 28.9 ± 11.9 | 23.4 |
| DANN | 53.1 ± 0.2 | 18.3 ± 0.1 | 44.2 ± 0.7 | 11.8 ± 0.1 | 55.5 ± 0.4 | 46.8 ± 0.6 | 38.3 |
| CDANN | 54.6 ± 0.4 | 17.3 ± 0.1 | 43.7 ± 0.9 | 12.1 ± 0.7 | 56.2 ± 0.4 | 45.9 ± 0.5 | 38.3 |
| MTL | 57.9 ± 0.5 | 18.5 ± 0.4 | 46.0 ± 0.1 | 12.5 ± 0.1 | 59.5 ± 0.3 | 49.2 ± 0.1 | 40.6 |
| SagNet | 57.7 ± 0.3 | 19.0 ± 0.2 | 45.3 ± 0.3 | 12.7 ± 0.5 | 58.1 ± 0.5 | 48.8 ± 0.2 | 40.3 |
| ARM | 49.7 ± 0.3 | 16.3 ± 0.5 | 40.9 ± 1.1 | 9.4 ± 0.1 | 53.4 ± 0.4 | 43.5 ± 0.4 | 35.5 |
| VREx | 47.3 ± 3.5 | 16.0 ± 1.5 | 35.8 ± 4.6 | 10.9 ± 0.3 | 49.6 ± 4.9 | 42.0 ± 3.0 | 33.6 |
| RSC | 55.0 ± 1.2 | 18.3 ± 0.5 | 44.4 ± 0.6 | 12.2 ± 0.2 | 55.7 ± 0.7 | 47.8 ± 0.9 | 38.9 |
| AND-mask | 52.3 ± 0.8 | 16.6 ± 0.3 | 41.6 ± 1.1 | 11.3 ± 0.1 | 55.8 ± 0.4 | 45.4 ± 0.9 | 37.2 |
| SelfReg | 58.5 ± 0.1 | 20.7 ± 0.1 | 47.3 ± 0.3 | 13.1 ± 0.3 | 58.2 ± 0.2 | 51.1 ± 0.3 | 41.5 |
| mDSDI | 62.1 ± 0.3 | 19.1 ± 0.4 | 49.4 ± 0.4 | 12.8 ± 0.7 | 62.9 ± 0.3 | 50.4 ± 0.4 | 42.8 |
| Fishr | 58.2 ± 0.5 | 20.2 ± 0.2 | 47.7 ± 0.3 | 12.7 ± 0.2 | 60.3 ± 0.2 | 50.8 ± 0.1 | 41.7 |
| ERM | 58.1 ± 0.3 | 18.8 ± 0.3 | 46.7 ± 0.3 | 12.2 ± 0.4 | 59.6 ± 0.1 | 49.8 ± 0.4 | 40.9 |
| ERM-NU (ours) | 60.9 ± 0.0 | 21.1 ± 0.2 | 49.9 ± 0.3 | 13.7 ± 0.2 | 62.5 ± 0.2 | 52.5 ± 0.4 | 43.4 |
| Mixup | 55.7 ± 0.3 | 18.5 ± 0.5 | 44.3 ± 0.5 | 12.5 ± 0.4 | 55.8 ± 0.3 | 48.2 ± 0.5 | 39.2 |
| Mixup-NU (ours) | 59.5 ± 0.3 | 20.5 ± 0.1 | 49.3 ± 0.4 | 13.3 ± 0.5 | 59.6 ± 0.3 | 51.5 ± 0.2 | 42.3 |
| SWAD | 66.0 ± 0.1 | 22.4 ± 0.3 | 53.5 ± 0.1 | 16.1 ± 0.2 | 65.8 ± 0.4 | 55.5 ± 0.3 | 46.5 |
| SWAD-NU (ours) | **66.6** ± 0.2 | **23.2** ± 0.2 | **54.3** ± 0.2 | **16.2** ± 0.2 | **66.1** ± 0.6 | **56.2** ± 0.2 | **47.1** |

Table 9: Results on DomainNet.

