# OpenReview forum: "Domain Generalization via Nuclear Norm Regularization"
_CPAL.cc/2024/Conference — CPAL 2024 (Proceedings Track) Oral_

### Official Review · Reviewer_odop · 2023-10-06
**Solid paper, may need additional empirical verification.**

**Rating:** 7
**Confidence:** 4

**Review:**

The paper proposes a simple yet effective method that leverages nuclear norm regularization to address the neural network's tendency to capture spurious correlations between labels and images instead of domain-invariant features. The authors provide a comprehensive set of experiments on synthetic and real datasets to showcase the effectiveness of their proposed method. Additionally, the inclusion of theoretical support in simpler settings improves the credibility of their approach.

## Pros and Cons
### Pros
The paper is clearly written and well-presented, all notations, and definitions are clearly conveyed, making it an enjoyable reading experience. Furthermore, most of the claims and conjectures mentioned in the paper are backed up using either empirical or theoretical results, enhancing the overall credibility of the research.

### Cons
The reviewer finds no major flaws in the paper. However, one aspect that raises some questions is the motivation presented in the paper: 'our main hypothesis is that environmental features have a lower correlation with the label than the invariant features.' While the theoretical analysis in Section 4 provides some support for this hypothesis, there is a lack of empirical evidence to reinforce this claim. Given that the setting in Section 4 is a simplistic setting, the reviewer believes that the intuition is not firmly established. Therefore, perhaps adding some experiments to empirically validate the motivation is useful. One starting point could be using the synthetic data introduced in Section 3.1 to visualize the magnitude of the matrix $A$ associated with environmental features and domain-invariant features, respectively, and see if the results meet the claims in the intuition.

## Questions

Q1: the notion $w$ is repeatedly used In line 259 when defining features and in line 284/285 to refer to the optimization variable (the entries of the classifier vector), which seems a bit confusing. The reviewer can understand that $w$ represents the strength of a corresponding feature, but why does it appear twice here?

Q2: In Proposition 3, the conclusion from the authors is that the OOD accuracy of ERM objective is "much worse than random guessing", while clearly, this is not the case in reality as we can observe from all the empirical results in the manuscript. Can the authors say more about the discrepancy between the theory and practice to explain this gap?

Q3: since the proposed method involves adding additional regularization on loss functions, the regularization strength becomes a tuning parameter. Do the authors conduct hyperparameter tuning for all tasks to find the suitable parameter?

---

### Official Review · Reviewer_sUq5 · 2023-10-07
**Simple approach with thorough evaluations.**

**Rating:** 6
**Confidence:** 3

**Review:**

Summary:

This paper proposes a nuclear norm regularizer to facilitate domain generalization. The proposed approach is evaluated on standard domain generalization benchmarks. Theoretical analysis with a toy example is given to justify the approach.

Advantages:

1. The paper is clearly written and easy to understand. The domain generalization problem is important and relevant.

2. The proposed regularizer is well-motivated with theoretical insights, and the synthetical data experiment offers clear intuition.

3. The method is thoroughly evaluated and compared with existing domain generalization baselines. Although the method itself is not the most performant approach, it shows compatibility with other approaches thanks to its simplicity.

4. The method is ablated carefully, giving the reader more insights into the method's property.

Downsides:

1. While the approach is simple (which is a commendable property, in my opinion), it can incur significant computation burden and instability, placing constraints on the batch size and the representation dimensionality.

2. The performance seems highly sensitive to the weighting factor $\lambda$, which may compromise the practicality of this method. Clearly, there is a subtle balance between norm regularization and classification performance, where the former, if not properly tuned, can throw away valuable information in the model representation.

---

### Official Review · Reviewer_Y5YN · 2023-10-10
**nuclear norm regularization**

**Rating:** 8
**Confidence:** 3

**Review:**

This paper studies domain generalization via nuclear norm regularization.

**Quality:**
The paper is technically sound with extensive experiments and some theoretical analysis. The method is simple but effective, and results are strong across diverse datasets.

**Clarity:**
The paper is well-written and easy to follow. The method and experiments are clearly explained.

**Originality:**
Applying nuclear norm regularization in this context is novel and provides a useful regularization technique for domain generalization.

**Significance:**
This technique could have a practical impact given the strong empirical results. The theory also provides new insights.

## Pros
1. Simple and efficient method that consistently improves over baselines.
2. Strong performance across a wide range of benchmark datasets.
3. Theoretical analysis offers insights into benefits of nuclear norm.
4. Easy to implement and combine with other methods.
5. No need for domain labels or changes to model architecture.

## Cons
1. Theoretical assumptions are restrictive; analysis is only for linear models.
2. Gains over existing methods are incremental, not a dramatic breakthrough.
3. Needs some hyperparameter tuning of regularization weight.
4. Sensitivity to hyperparameters could be analyzed more.
5. More analysis to connect theory to deep networks would be beneficial.

---

### Meta-Review · Area_Chair_dkYz · 2023-11-13

**Recommendation:** Accept (Poster)
**Confidence:** 5

**Metareview:**

This paper proposes nuclear norm regularization as a method for domain generalization, and provides empirical and theoretical justifications. The reviewers appreciate the contributions of this paper and unanimously recommend acceptance. One issue pointed out by the reviewers is the sensitivity to hyperparameters, which the authors acknowledge in their response.

---

### Decision · Program_Chairs · 2023-11-19

**Decision:**

Accept (Oral)

**Comment:**

The paper tackles the important problem domain generalization using a novel nuclear norm regularization, with both theoretical and empirical justifications. The writing of the paper is clear and easy-to-follow.

The action PC chair for this paper is Yuejie Chi, who made the decision after carefully reading the paper as well as the comments by all reviewers and AC. The decision is agreed by all PC chairs.